# Effect of Dopamine on Growth, Some Biochemical Attributes, and the Yield of Crisphead Lettuce under Nitrogen Deficiency

**Saad Farouk** [1,*], **Mahmoud A. M. Abd El-Hady** [2], **Mohamed A. El-Sherpiny** [3], **Mohamed M. Hassan** [4], **Khalid H. Alamer** [5], **Sami Asir Al-Robai** [6], **Esmat F. Ali** [4,*] **and Hemat A. El-Bauome** [7]

1    Agricultural Botany Department, Faculty of Agriculture, Mansoura University, Mansoura 35516, Egypt
2    Horticulture Department, Faculty of Agriculture, Damietta University, Damietta 34517, Egypt;
     m_abdelhady@du.edu.eg
3    Water and Environment Research Institute, Agriculture Research Centre, El-Gama St., Giza 12619, Egypt;
     m_elsherpiny2010@yahoo.com
4    Department of Biology, College of Science, Taif University, P.O. Box 11099, Taif 21944, Saudi Arabia;
     m.khyate@tu.edu.sa
5    Biological Sciences Department, Faculty of Science and Arts, King Abdulaziz University,
     Rabigh 21911, Saudi Arabia; kalamer@kau.edu.sa
6    Department of Biology, Faculty of Science, Al-Baha University, Al-Baha P.O. Box 1988, Saudi Arabia;
     salrobai@bu.edu.sa
7    Vegetable and Floriculture Department, Faculty of Agriculture, Mansoura University, Mansoura 35516, Egypt;
     haelbauome@mans.edu.eg
*    Correspondence: gadalla@mans.edu.eg (S.F.); a.esmat@tu.edu.sa (E.F.A.)

**Abstract:** Nitrogen (N) represents the most important nutrient for plant growth and productivity, but extreme and ineffective usage of N fertilizer results in boosted plant production expenditures and environmental contamination. For the world's sustainable food production and environmental profits, there has been increased research interest in reducing the use of N fertilization along with improving plant N deficiency (ND) tolerance. Dopamine (DA), a potential antioxidant, mediates several physio-biochemical processes in plants under normal or stressful conditions. However, their roles in increasing ND tolerance in crisphead lettuce are not well-documented. We investigate the role of DA concentration (0.50 and 100 μM) on the growth and yield of crisphead lettuce plants under ND. Under normal conditions (100% recommended N fertilizer dose), DA (50 and 100 μM) application significantly enhanced growth, chlorophyll concentration, N%, antioxidant enzymes activity, as well as yield and its components, decreased nitrate accumulation and oxidative biomarkers compared to untreated plants (0 μM DA). ND significantly decreased plant growth and yield attributes as well as evoked oxidative impairment and nitrate accumulation as compared to 100% recommended N fertilizer dose in the absence of DA. However, within ND conditions, the application of DA concentrations significantly mitigated ND-induced oxidative burst and improved plant growth, chlorophyll concentration, N%, nitrate concentration, peroxidase, catalase, superoxide dismutase, total soluble solid, vitamin C, dry matter %, and total sugars, over 0 μM DA treated plants. Current findings highlighted that exogenous application of 100 μM DA could reinforce the crisphead lettuce plant's resilience to ND by minimizing reactive oxygen species accumulation and promoting enzymatic antioxidants alongside growth, yield, and quality improvement. The beneficial effects of DA in lessening ND's drastic impacts on crisphead lettuce resulted from upregulating antioxidant enzyme activity, impairment of oxidative biomarkers, and maintaining chlorophyll levels. The current findings open pioneering prospects to reduce nitrogen fertilization by DA application without any drastic effect on plant productivity. But further research is needed to fully understand DA effects and their mechanisms in inducing ND tolerance in different plant species, including crisphead lettuce.

**Keywords:** crisphead lettuce; dopamine; nitrogen deficiency; oxidative biomarker; quality

## 1. Introduction

Nitrogen (N) is an essential constituent of plant protein, nucleotides, enzymes, chlorophyll, ATP, and phytohormones, which plays a crucial role in photosynthesis, respiration, as well as protein and fat metabolism [1,2]. Also, N improves root architecture (volume, area, diameter, and dry mass) alongside increasing nutrient uptake and dry mass production [2,3]. Alternatively, excessive N fertilizer supplementation is expensive and hastens the deprivation of the environment and human health [2,4]. Due to the fact that N is elaborate in most plant metabolic pathways, nitrogen deficiency (ND) is able to influence photo-assimilation allocation arrays, alter serious physio-biochemical pathways and molecular reactions, plant establishment, stunted growth, and reduced flowering and seed setting [1,5,6]. Since more than half of the N in leaves is devoted to chloroplasts, ND also accelerates leaf chlorosis and experiences a great reduction in photosynthetic rates that diminish plant development and marketable values of leafy plants [5–7]. More importantly, ND accelerates the over-production of reactive oxygen species (ROS) that causes ubiquitous cellular destruction and dysfunction of numerous physio-molecular responses [7,8]. Within oxidative stress, crops possess a distinct protection approach for eradicating ROS and maintaining redox homeostasis [9–11]. As a result, boosting ND tolerance is an effective strategy for lowering nitrogen leakage and environmental pollution caused by excessive N fertilization.

Dopamine (DA) is a class of catecholamine substances and a biogenic amine (with a3, 4-dihydroxy-substituted phenyl ring) that is found in many plant species [5,12]. Extra attention has been paid to DA in crop production since its recognition as an imperative hormone and neurotransmitter in mammals [13,14]. In contrast to the enormous information on their function in mammals, their roles in plants are currently less well understood. The DA plays as a novel class of regulatory molecules in plants since it was first recognized as a strong plant water-soluble antioxidant with a superior antioxidative aptitude than catechin, glutathione, flavone, luteolin, and flavonol quercetin as well as similar potency to gallocatechin gallate and ascorbic acid [12,13]. It regulates sugar metabolism, improves nutrient uptake, increases photosynthetic capacity and photophosphorylation, harmonizes plant hormones, maintains water status, and increases chlorophyll concentration, alongside promoting plant growth under normal or stress conditions, including ND [3,5,12,15]. Moreover, DA controls the expression of several stress-associated genes, which highlights its function as multi-regulatory molecules [3,15,16]. It also induces plant stress tolerance through its roles and its derivative melanin as a powerful free radical scavenger and maintains redox homeostasis [13,14]. Therefore, DA may help plants to resist ND through antioxidant regulation; however, the number of investigations on its roles in moderating the drastic impact of ND on crisphead lettuce is still low [3,4]. The available evidence is rarely contradictory, depending on the plant species and application time.

Lettuce consumption is extensively endorsed, as lettuce offers innumerable benefits to human health owing to its high levels of bioactive phytochemical compounds [1]. It has a high-water percentage (about 95%) and is low in calories [17]. Additionally, it is a great source of vitamins, minerals, and bioactive substances, including polyphenols, flavonoids, and natural pigments, all of which have positive health effects [17–19]. Consuming leafy vegetables is concomitant with a minor risk of chronic diseases, including cancer, Alzheimer's, diabetes, and cardiovascular disease [20,21]. Additionally, it is a good source of vitamin A, which supports eye health, and vitamin K, which is needed for bone health and mitigating blood clotting [17,22]. Natural plant antioxidant molecules have become a widespread tendency owing to the overall awareness of the significance of a healthy diet and restrictions on the usage of harmful artificial antioxidants [23]. Crisphead lettuce (*Lactuca sativa* var. capitate) is a type of lettuce that forms a tight, compact head of leaves.

The current study proposes to assess the effect of DA on growth, oxidative biomarkers, antioxidant capacity, and nitrate accumulation in ND-affected crisphead lettuce plants. We hy-

pothesize that DA spraying would enhance crisphead lettuce plants to withstand ND through strengthening antioxidant enzyme activities alongside impairment of oxidative biomarkers.

## 2. Materials and Methods

### 2.1. Experimental Layout

A field trial was carried out at the experimental farm of Mansoura University, Egypt (31°22′40.3″ E longitude and 31°02′40.6″ N latitude, altitude 15 m above sea level) under a furrow irrigation system, to assess the role of N fertilizer levels (100% or 50% recommended N 'RN', that introduced by the Ministry of Agriculture and Land Reclamation, Egypt; i.e., 144 or 72 N unit ha$^{-1}$ as ammonium sulfate, 20.5% N) and foliar application of DA concentrations (0, 50, 100 μM) on crisphead lettuce "*Lactuca sativa* L. cv Kharga" growth, some physiological and yield attributes. The chemical characteristics of the initial soil before transplanting are estimated according to the Sparks et al. [24] protocol and presented in Table 1.

**Table 1.** Some chemical characteristics of initial experimental soil traits (at a depth of 0.0–30 cm).

| Chemical Parameters | Electric Conductivity (EC) | pH | Organic Matter (OM) |
|---|---|---|---|
| | 3.49 (dSm$^{-1}$) | 8.00 | 1.39 (%) |
| Available nutrients (mg kg$^{-1}$) | Nitrogen | Phosphorus | Potassium |
| | 48.0 | 7.99 | 235.8 |

A field experiment was performed following a randomized complete design, with three replicates, including six treatments as follows:

T1, 100% RN without DA
T2, 100% RN with 50 μM DA
T3, 100% RN with 100 μM DA
T4, 50% RN without DA
T5, 50% RN with 50 μM DA
T6, 50% TN with 100 μM DA

One month prior to transplanting, farmyard manure and calcium super-phosphate (15.5% $P_2O_5$) were applied simultaneously at a rate of 15.0 t ha$^{-1}$ and 720 kg ha$^{-1}$, respectively. The experimental field was divided into 18 plots, each one containing three rows (9 m long and 0.85 m wide), then irrigation. Crisphead lettuce "*Lactuca sativa* L. cv Kharga" seedlings (21 days old) were transplanted on 19 Dec 2022 in the middle of the ridges with a spacing of 40 cm among plants. Nitrogen fertilization (100% or 50% RN) was divided into two equal doses for each N level and then added at 14 and 28 days from the transplanting. Potassium sulfate (48% $K_2O$) was applied concurrently with the N at a rate of 60 units $K_2O$ ha$^{-1}$. Foliar DA (3,4 Dihydroxyphenethylamine Hydrochloride, FUJIFILM Wako Pure Chemical Corporation; Richmond, VA, USA) spraying was performed five times at 20, 30, 40, 50, and 60 days after transplant.

### 2.2. Sampling

At 75 days from transplanting, nine lettuce plants (each three plants representing one replicate) from each replicate were taken for measuring some morpho-physiological and yield attributes.

### 2.2.1. Vegetative Growth and Yield Attributes

Total leaf area (cm$^2$), plant fresh weight (kg plant$^{-1}$), number of outer leaves per plant, head weight (kg plant$^{-1}$), and total yield (t ha$^{-1}$) were measured. Total leaf area plant$^{-1}$ was calculated using a subsequent equation following Koller [25] method.

$$\text{Leaf area plant}^{-1} \text{ (cm}^2) = \frac{Disk\ area\ of\ 10\ disks\ from\ inner\ and\ outer\ leaves \times fresh\ weight\ of\ leaves}{Fresh\ weight\ of\ 10\ disks}$$

### 2.2.2. Photosynthetic Pigments and Chemical Constituents

With ice-cold methanol, chlorophyll (Chlorophyll a, Chlorophyll b, and total chlorophyll) were extracted from the outer leaves for 2 days at lab. temperatures in the dark and quantified spectrophotometrically (T60 UV-Visible spectrophotometer, PG Instrument Limits, Lutterworth, UK) as described by Lichtenthaler and Wellburn [26].

For N determination, 0.2 g fine dry powdered leaves were carefully transferred to a digestion flask with 5 mL of $H_2SO_4$ and heated at 100 °C for 2 h; subsequently, an aliquot of $H_2SO_4/HClO_3$ mixture was decanted dropwise; then the digestible was cooled for 15 min, and then determined N using Micro-Kjeldahl method [27].

The lettuce plant's nitrate concentration was determined in acetic acid extract using N-1naphthyle ethylene diamine dihydrochloride as an indicator. The intensity of the pink color of the filtrate was measured at a wavelength of 450 nm using a spectrophotometer, following the procedure described by Singh [28].

### 2.2.3. Malondialdehyde, Hydrogen Peroxide, and Superoxide Anion

The quantification of malondialdehyde (MDA) in lettuce leaves was achieved using the thiobarbituric acid reactive substances (TBARS) method, as described by Heath and Packer [29]. Initially, lettuce plant tissue was homogenized in a buffer solution, followed by the addition of trichloroacetic acid (TCA) to precipitate proteins. Afterward, the resulting supernatant was treated with thiobarbituric acid (TBA), which reacted with MDA to generate a pink chromophore, which was then measured spectrophotometrically at 532 nm wavelength.

The concentration of hydrogen peroxide ($H_2O_2$) in lettuce plant samples was determined spectrophotometrically using the titanium-peroxide method. This method involves the reaction of $H_2O_2$ with titanium(IV) ions in acidic conditions to produce a yellow-colored complex. The intensity of the yellow color is directly proportional to the $H_2O_2$ concentration and can be measured spectrophotometrically at a wavelength of 410 nm, as described by Tariq et al. [30].

The concentration of superoxide anion ($O_2^{\bullet-}$) in lettuce plant samples was determined using Mohammadi and Karr [31] protocol. Superoxide anion was determined by incubating the leaf tissues in an extraction solution made up of 20 mM sodium phosphate buffer, 20 μM nicotinamide adenine dinucleotide (NADH), and 100 μM ethylenediaminetetraacetic acid (EDTA) disodium salt in hermetically sealed tubes. The reaction was ongoing with the addition of 25.2 mM epinephrine. Samples were incubated at 28 °C, remaining under shaking for 5 min, then reading of absorbance at 480 nm.

### 2.2.4. Antioxidants Enzymes

The standard procedures outlined by Elavarthi and Martin [32] were used to measure the enzymatic antioxidant activity. The activity of the peroxidase enzyme activity (POD, unit $min^{-1}$ $g^{-1}$ protein) was estimated by recording the oxidation of guaiacol at 470 nm using a spectrophotometer. The reaction mixture contained guaiacol, $H_2O_2$, and crude plant extract. The rate of change in absorbance was measured over time.

Catalase enzyme (CAT) activity was assessed by recording the breakdown of $H_2O_2$ at 240 nm using a spectrophotometer. The reaction mixture contained $H_2O_2$ and crude plant extract. The rate of change in absorbance was measured over time, and the CAT activity was expressed as a unit $min^{-1}$ $g^{-1}$ protein.

Superoxide dismutase (SOD) enzyme activity was determined using the nitroblue tetrazolium (NBT) assay. The reaction mixture contained NBT, riboflavin, and crude plant extract. The reduction of NBT by superoxide radicals was inhibited by SOD, and the rate of change in absorbance was measured over time. The SOD activity was expressed as a unit $min^{-1}$ $g^{-1}$ protein.

2.2.5. Head Quality

Total sugar, total dissolved solid (TDS), vitamin C, and dry matter % were estimated according to AOAC [33].

*2.3. Statistical Analysis*

The data homogeneity was achieved formerly before the analysis of variance (ANOVA). All data were analyzed via a one-way ANOVA with the COSTATC statistical program (CoHort software, release 6.3.0.3, 2006; NC, USA) to test the impact of DA on moderating ND injury. The Tukey's Honestly Significant Difference (HSD) test was used for comparing means at $p \leq 0.05$. The means and standard deviation (SD) of three independent biological replications were utilized to present the data

# 3. Results

*3.1. Vegetative Growth Criteria*

Crisphead lettuce plants' growth was greatly decreased by ND after 75 days from transplanting; however, growth was restored with DA spraying. Commonly, ND significantly ($p < 0.05$) reduced leaf area (Figure 1a) by 18.96%, fresh weight (Figure 1b) by 28.03%, number of outer leaves (Figure 1c) by 28.61%, head weight (Figure 1d) by 38.17%, and total yield (Figure 1e) by 38.21% compared with a full RN. Application of DA at both levels significantly ($p < 0.05$) boosted crisphead lettuce plant growth over non-treated plants (0 µM DA). Spraying with 100 µM DA increased leaf area (Figure 1a), fresh weight (Figure 1b), number of outer leaves (Figure 1c), head weight (Figure 1d), and total yield (Figure 1e) by 5.41, 14.01, 22.97, 17.57, and 17.62% above control plants '100% RN + 0 µM DA'. A parallel trend was achieved within ND; crisphead lettuce plants treated with DA displayed a superior growth than no DA under ND-influenced plants (Figure 1a–e). Application of 100 µM DA was extra effective compared to 50 µM DA in moderating the destructive impact of ND on crisphead lettuce plant's leaf area (Figure 1a), fresh weight (Figure 1b), number of outer leaves (Figure 1c), head weight (Figure 1d), and total yield (Figure 1e) that deliberated substantial rises that were 7.17, 33.76, 25.07, 25.15, and 25.19% relative to ND-affected plants only.

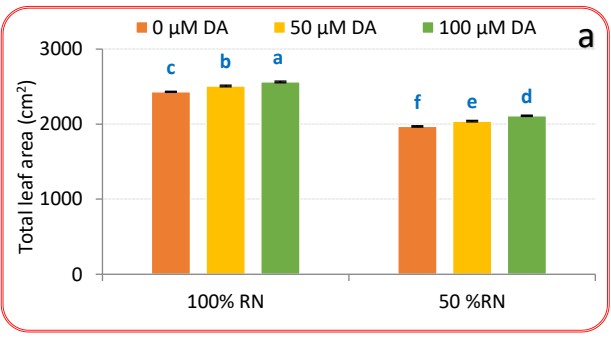

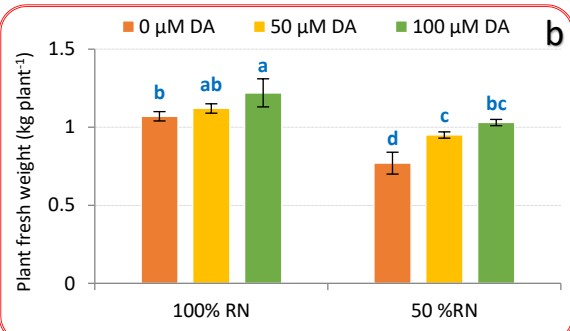

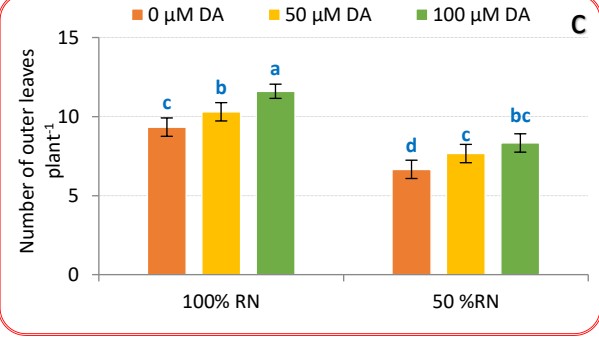

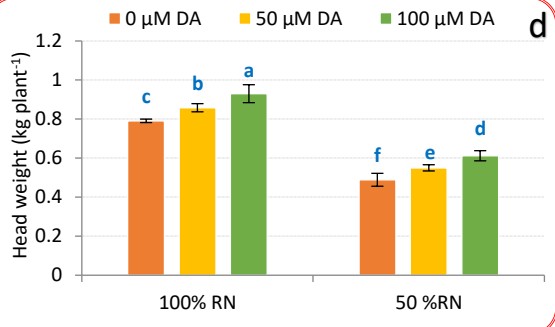

**Figure 1.** *Cont.*

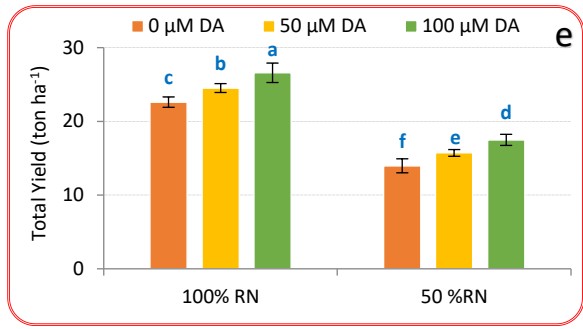

**Figure 1.** Effect of dopamine (DA) and nitrogen levels (RN) on certain growth characteristics ((**a**), Total leaf area; (**b**), Plant fresh weight; (**c**), number of outer leaves; (**d**), Head weight; (**e**), total yield) of crisphead lettuce plants. Data represent means ± SD of three replicates. Means within columns followed by different letters are significantly different ($p \leq 0.05$).

### 3.2. Photosynthetic Pigments, Nitrogen% and Nitrate Concentration

Chlorophyll concentration (Figure 2a–c), N percentage (Figure 2d), and nitrate concentration (Figure 2e) were significantly reduced by 50% RN proportionate to 100% RN. DA spraying under 100% or 50% RN fertilizers significantly raised the concentration of chlorophyll a, chlorophyll b, total chlorophyll, and N percentage while decreasing nitrate accumulation relative to untreated crisphead lettuce plants under such N levels. Spraying with 100 µM, DA deliberated the best outcomes that increase the chlorophyll a, chlorophyll b, total chlorophyll, and N percentage by 3.23, 6.06, 4.14, 6.42%, or by 5.53, 7.51, 6.20, 7.60%, respectively, corresponding to plants supplemented with 100% or 50% RN without DA application, respectively. Moreover, Figure 2e also designates that the application of DA decreased nitrate accumulation by 5.99% and 11.54% as compared to non-treated plants with DA under 100% or 50% RN.

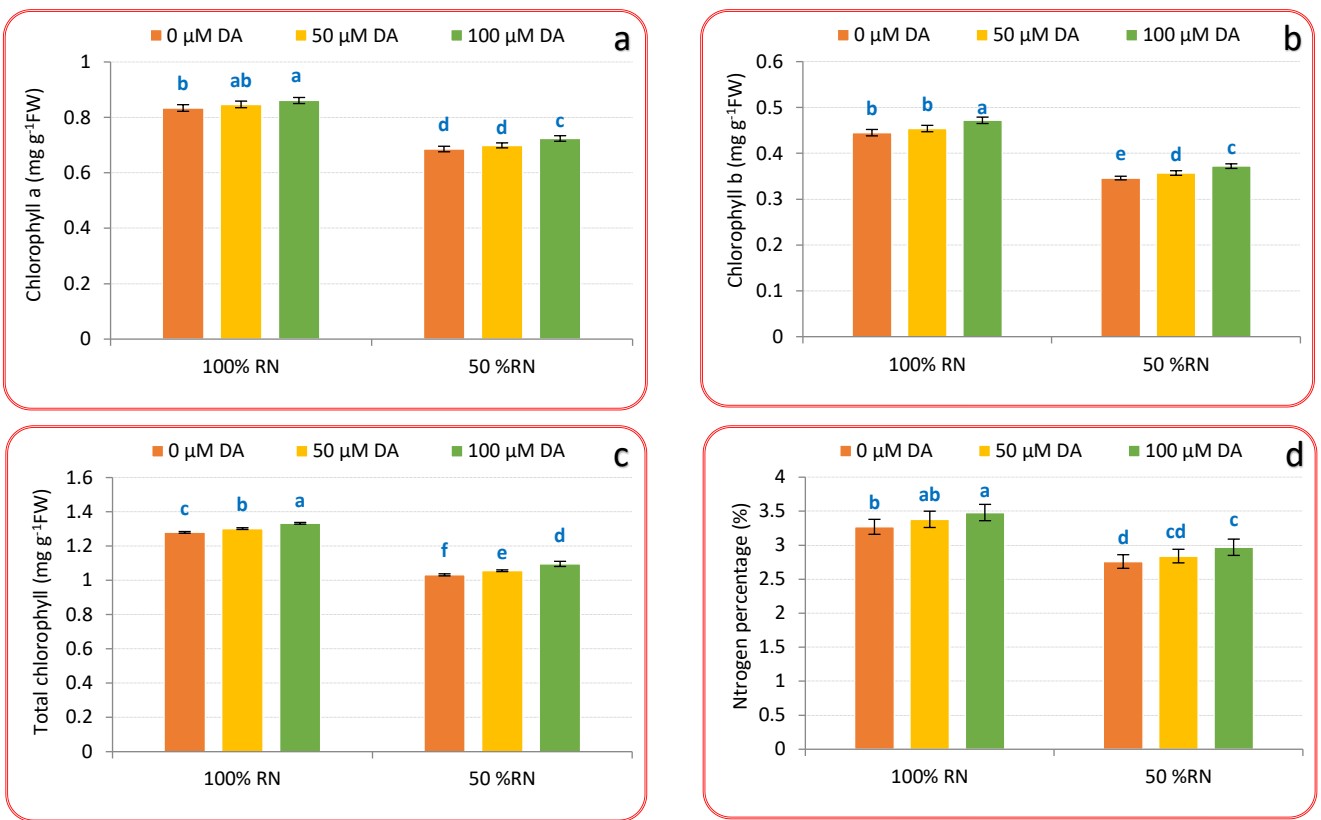

**Figure 2.** *Cont.*

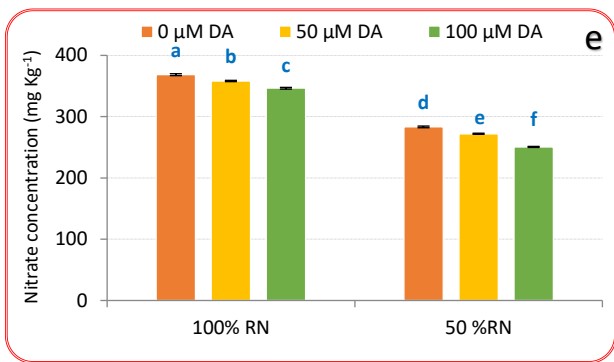

**Figure 2.** Effect of dopamine (DA) and nitrogen levels (RN) on chlorophyll a (**a**), chlorophyll b (**b**), total chlorophyll (**c**), nitrogen (**d**), and nitrate (**e**) level of crisphead lettuce plant at harvest stage. Data represent means ± SD of three replicates. Means within columns followed by different letters are significantly different ($p \leq 0.05$).

### 3.3. Oxidative Biomarkers

For assessment of the effect of DA in alleviating ND-induced oxidative injuries, the accumulation of $H_2O_2$, MDA, and $O_2^{\bullet-}$ was assessed (Figure 3a–c). ND prompted a substantial accumulation of $H_2O_2$, MDA, and $O_2^{\bullet-}$ above control plants (100% RN), that increased by 80.72, 51.99, and 88.77%, respectively, within the ND condition. Under 100% RN, DA spraying (specifically 100 μM) significantly declined $H_2O_2$, MDA, and $O_2^{\bullet-}$ relative to 100% N. Relative to ND-affected plants only, DA spraying significantly ($p < 0.05$) deteriorated the oxidative biomarkers in crisphead lettuce plants. Amongst DA levels, 100 μM deliberated the supreme lessening in oxidative biomarkers than 50 μM DA, which reduced $H_2O_2$ (62.00 and 51.00%), MDA (27.31 and 10.66%), and $O_2^{\bullet-}$ (42.85 and 13.26%) compared with 0 μM DA and ND-affected plants (Figure 3a–c). Current findings recognized that the DA spraying might abolish ND-evoked ROS assimilation and the consequent oxidative burst in crisphead lettuce plants.

### 3.4. Antioxidants Enzymes

The findings in Figure 4a–c indicate that both concentrations of DA significantly increased the activity of antioxidant enzymes compared to those of untreated plants under 100% or 50% RN. Furthermore, it can be observed that the values of enzymatic antioxidants were lower with 100% RD compared to 50% RN. Corresponding to untreated 100% RN, the foliar spraying with 50 or 100 μM DA enhanced the antioxidant enzyme activity in crisphead lettuce plants. The uppermost activities of POD (Figure 4a), CAT (Figure 4b), and SOD (Figure 4c) were achieved under 100 μM DA over non-treated control plants. It is manifest that ND persuades substantial increases in POD (Figure 4a), CAT (Figure 4b), and SOD (Figure 4c) activity over their relevant control plants without DA application. The application of DA to ND-influenced plants additionally enhanced POD, CAT, and SOD activities relative to ND-influenced plants only. The maximum values of POD (Figure 4a), CAT (Figure 4b), and SOD (Figure 4c) were achieved under the treatment of 50% RN with DA (100 μM), while the minimum values were observed under the treatment of 100% RN without DA. The findings also indicate that both concentrations of DA significantly increased the values compared to those of untreated plants, as the values increased with the DA rate under both N doses. Furthermore, it can be observed that the values of enzymatic antioxidants were lower with 100% RN compared to 50% RN. These outcomes revealed that DA might have vital meanings in antioxidant systems that can be elaborate in moderating ND-evoked oxidative injury in crisphead lettuce plants.

### 3.5. Head Quality

Both N level and DA concentration significantly affected quality traits, such as total sugar (Figure 5a), total dissolved solids (TDS) (Figure 5b), vitamin C (Figure 5c), and dry

matter (Figure 5d) of the crisphead lettuce plant. The findings indicate that the maximum quality values were achieved due to the treatment of 100% RN with DA (100 μM), while the minimum values were recorded within 50% RN without DA. Also, it can be noticed that both concentrations of DA significantly increased the quality values compared to that of the untreated plants with DA, as the values increased as the rate of DA increased under 100% or 50% RN. Moreover, the quality values were greater with the 100% RN than that with 50% RN. On the other hand, it can be noticed that the treatment of 50% RN with DA (100 μM) achieved results close to those achieved with the treatment of 100% RN without DA.

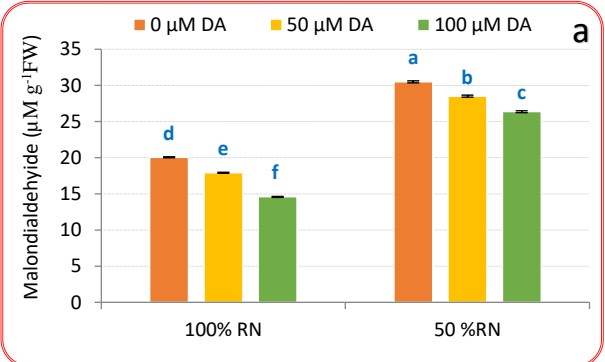
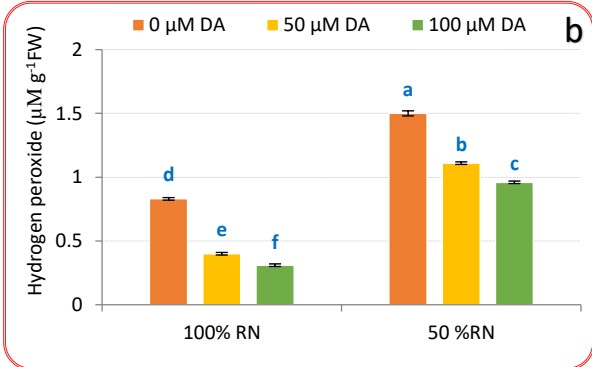

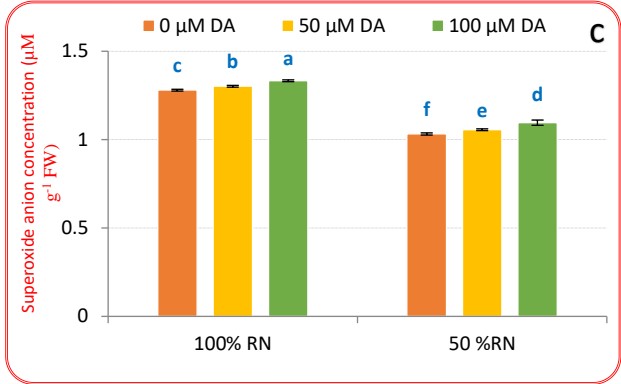

**Figure 3.** Effect of dopamine (DA) and nitrogen levels (RN) on markers for oxidative stress ((**a**), Malondialdehyide; (**b**), hydrogen peroxide; (**c**), superoxide anion) of crisphead lettuce plant at harvest stage. Data represent means ± SD of three replicates. Means within columns followed by different letters are significantly different ($p \leq 0.05$).

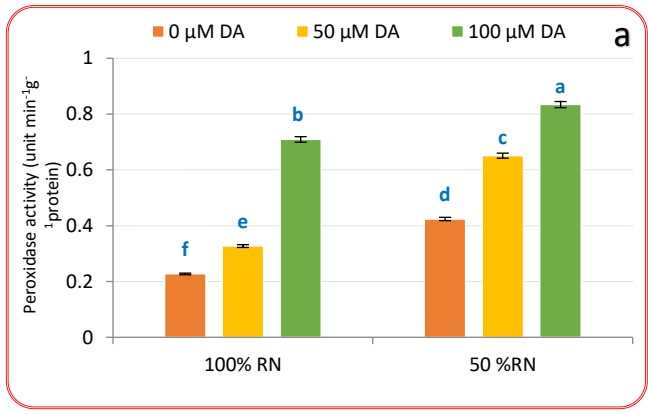
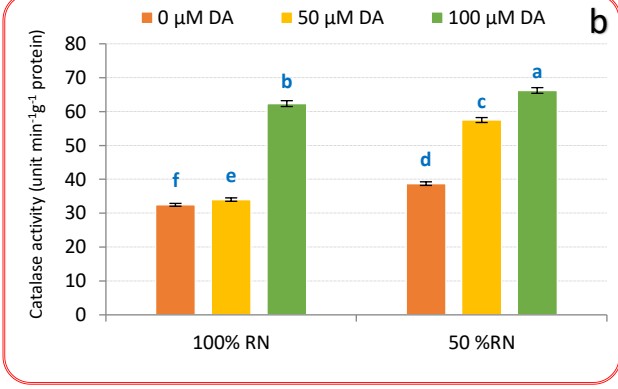

**Figure 4.** *Cont.*

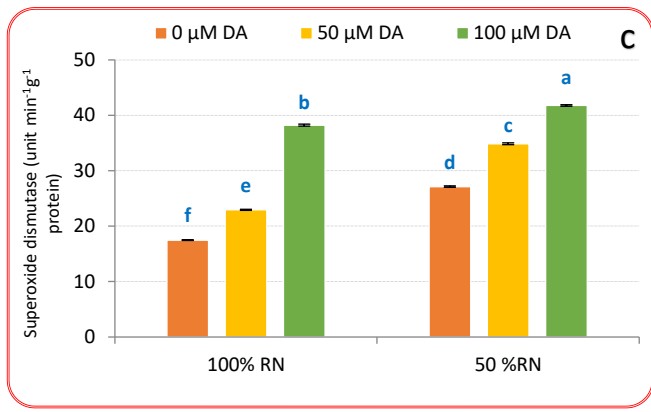

**Figure 4.** Effect of dopamine (DA) and nitrogen levels (RN) on the activity of some antioxidant enzymes ((**a**), Peroxidase; (**b**), catalase; (**c**), superoxide dismutase ) of crisphead lettuce plant at harvest stage. Data represent means ± SD of three replicates. Means within columns followed by different letters are significantly different ($p \leq 0.05$).

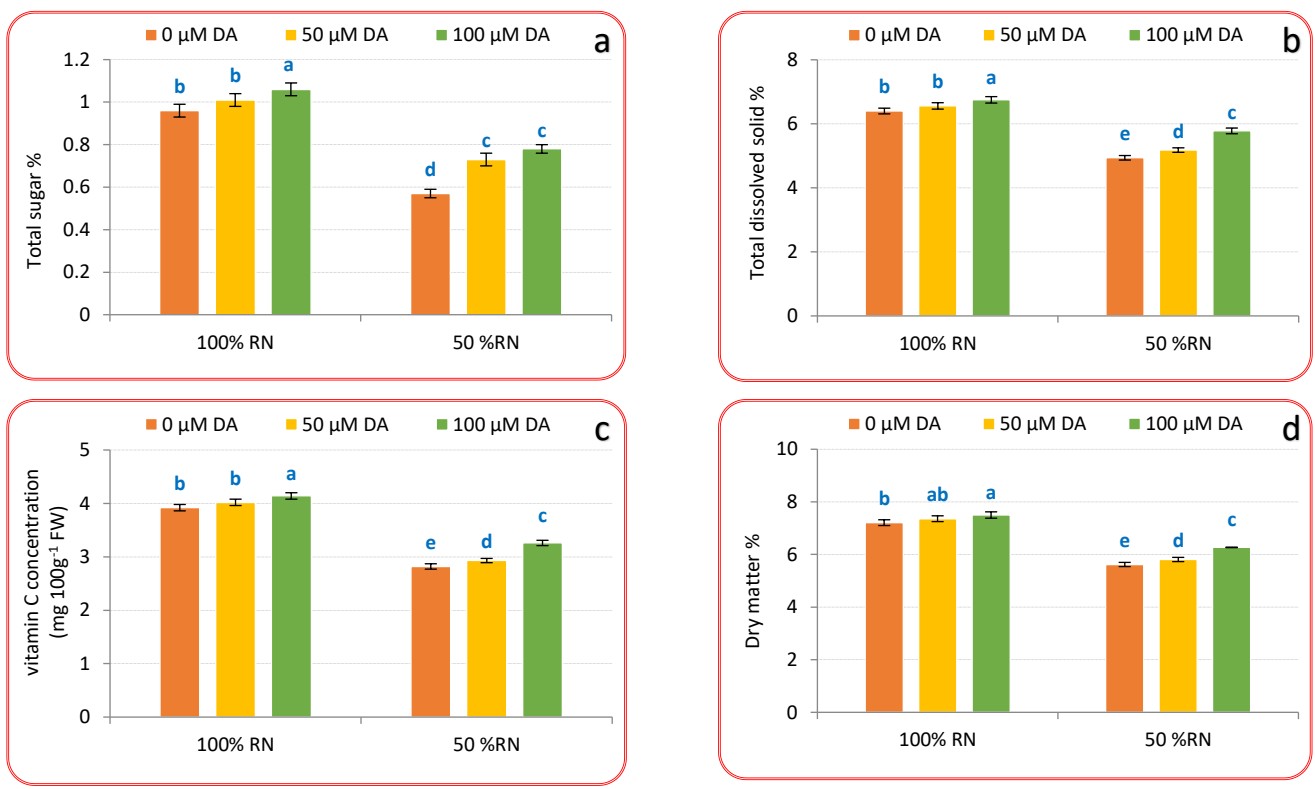

**Figure 5.** Effect of dopamine (DA) and nitrogen levels (RN) on some quality parameters ((**a**), Total sugar; (**b**), Total dissolved solid; (**c**), Vitamin C; (**d**), dry matter percentage) of crisphead lettuce plant at harvest stage. Data represent means ± SD of three replicates. Means within columns followed by different letters are significantly different ($p \leq 0.05$).

*3.6. Pearson Correlations and a Heat Map*

The exploration of Pearson's correlation was made to perceive the relationship among the parameter achieved in ND-crisphead lettuce sprayed with DA (Figure 6a). The outcomes exhibited a remarkably positive correlation ($p \leq 0.05$) between leaf area, plant fresh weight, outer leaf number per plant, head weight, total yield, chlorophyll a, chlorophyll b, total chlorophyll, N%, nitrate concentration, total sugar, total soluble solid, vitamin C concentration, and dry matter % with a coefficient correlation value close to 1. Meanwhile, they were negatively correlated ($p \leq 0.05$) with MDA, $H_2O_2$, $O_2^{\bullet-}$, POD, CAT, and SOD.

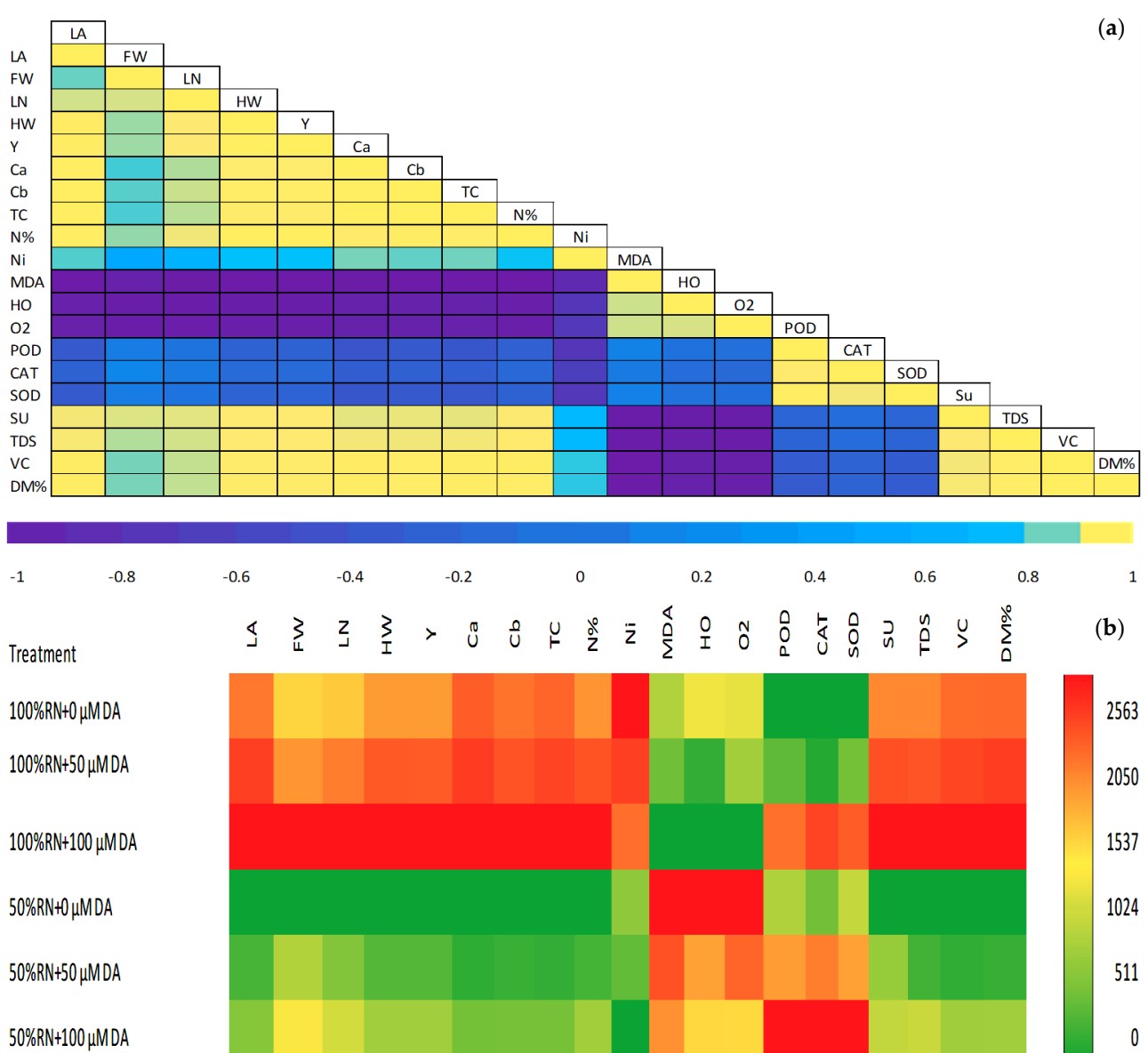

**Figure 6.** Pearson correlations and heat map. (**a**) Graph of Person's correlated exploration amongst the trials studied. The colors exemplify dissimilarities in the resulting data. Yellow designates a positive correlation, and purple designates a negative correlation; (**b**) in the heatmap graph, the colors exemplify dissimilarities in the resulting data. LA, leaf area; FA, plant fresh weight; LN, leaves number per plant; HW, head weight; Y, total yield; Ca, chlorophyll a; Cb, Chlorophyll b; TC, total chlorophyll; N%, N percentage; Ni, nitrate concentration; MDA, Malondialdehyde; HO, hydrogen peroxide; $O_2$, superoxide; POD, peroxidase; CAT, catalase; SOD, superoxide dismutase; Su, total sugars; TDS, total soluble solid; VC, vitamin C; DM%, dry matter percentage.

Along with the correlation exploration, the relationship between the parameters deliberated and DA foliar spraying for ND-stressed crisphead lettuce plants was examined using the heatmap. As shown in Figure 6b, leaf area, fresh weight, the number of outer leaves, head weight, total yield, chlorophyll a, chlorophyll b, total chlorophyll, N%, total sugars, total dissolved solids, vitamin c, and dry matter % were reduced (dark green) by ND, but enhanced by DA concentration (light green, yellow or red). In contrast, MDA, $H_2O_2$, and $O_2^{\bullet-}$ were increased by ND stress (dark red) but decreased with DA concentration. The DA at 100 showed the best results in reducing oxidative biomarkers and increased plant growth, yield, and quality under 100 RN or 50 RN. Finally, the activity of antioxidant

enzymes increased under 50 RN compared to 100 RN. Additionally, the application of DA concentration increased their activities under 100 or 50% RN overnon-treatedd plants.

## 4. Discussion

Current findings stated that ND causes a declining effect on crisphead lettuce plant growth and productivity. Alternatively, DA application reinforces plant productivity under ND by minimizing ROS accumulation and promoting antioxidant enzyme activity. These outcomes were constant with earlier studies which indicate that the DA effect was dose and time-dependent [3,5,34].

The current study found that crisphead lettuce plant growth, productivity, and quality were significantly reduced within ND, but the DA application under 100% or 50% RN significantly boosted all growth and yield attributes (Figures 1 and 5). The role of DA in lessening ND injuries on crisphead lettuce plants is, however, not previously documented. Nitrogen, along with carbon, represents the most important nutrient for leafy crop growth and productivity, as well as constituting the majority of plant biomass [35]. So, under ND, plant growth and productivity were dramatically reduced, as reported previously [3,5–7]. This decrease is frequently attributed to that ND-persuaded malady in almost physio-molecular changes, i.e., hindrance of water and photo-assimilates translocation; destruction of plant photosynthetic efficiency and damage of ATP assimilation prerequisite for plant establishment [5,7,36]; hampered ion uptake and similarly, the extreme ROS creation [3,8]. Since N is the synthesis of several enzymes and chlorophyll required for photosynthesis, ND certainly upsets $CO_2$ assimilation and a deterioration in photosynthetic rates, alongside a decline in photoassimilate translocation and biomass accumulation, leading to a decline in plant growth and productivity [5,7,37]. Exogenous DA application significantly increased plant growth and yield attributes under 100% or 50% RN, and 100 μM DA was the utmost effective level in this regard. These outcomes revealed that DA is able to lessen the inhibition of growth and yield within ND. Accordingly, Jiao et al. [38] revealed that the DA application significantly increased plant fresh weight, dry weight, and the number of leaves per plant. The mitigating effects of DA on the growth and yield of crisphead lettuce plants within ND were directly or indirectly ascribed to the role of DA in nutrient uptake and root system architecture [12,34], photosynthetic capacity and pigments [15,38], stomatal regulation [5], N fixation, and photophosphorylation [14]. Furthermore, in the current study, the possible motivation for plant growth by DA supplementation might be attributable to depressed oxidative destruction to ND-affected plants (Figure 3) via enhancing the antioxidant system (Figure 4) and is efficient in scavenging ROS [12,14,34]. The relationship between DA and other hormones in plants has already been discussed; in this concern, DA might prevent the deprivation of indole acetic acid and sustain a greater concentration of auxin in plant tissues [39]. Additionally, Kamisaka [40] established that catecholamines and their by-products boosted gibberellic acid production on lettuce hypocotyl. Moreover, Lan et al. [15] found that the application of DA on N -affected cucumber plants significantly increased photosynthetic rate, gas exchange, and intercellular $CO_2$ concentration, contributing to improving photosynthetic processes and producing more photoassimilate alongside improving plant biomass accumulation and plant productivity.

Chlorophylls are crucial plant pigments that absorb light energy and transport electrons to the reaction center throughout photosynthesis pathways [35]. In the present investigation, chlorophyll concentration significantly decreased under ND; however, exogenous DA treatment clearly reversed these negative effects. These outcomes were approved by former studies [5,6,34,36]. The declining impacts of ND on chlorophyll levels may be a result of the switch-off chlorophyll assimilation genes, resulting in the diminishing of ALA synthesis pathways and chlorophyll decline [7,16,36]. Accordingly, Chen [41] found that stress factors upregulate chlorophyll synthesis genes, including MdHEMB1 and MdHEMB2, which encode ALAD. Moreover, earlier studies have advocated that the overproduction of ROS is the main cause of chlorophyll decrease under stress conditions (Figure 3; [42]). Exogenous DA significantly increased chlorophyll concentration under 100

or 50% RN, which designated that DA lessened the deleterious effect of ND on chlorophyll and that 100 μM DA was the most effective concentration. Accordingly, Lan et al. [15] found that the application of DA on ND-affected cucumber plants considerably raised the concentration of chlorophyll a and chlorophyll b. Xiao-min et al. [5]; found that the application of DA significantly increased chlorophyll a, chlorophyll b, and total chlorophyll in apple leaves under 100 or 50% of RN. Jiao et al. [14] found that the application of DA significantly increased chlorophyll a, chlorophyll b, and total chlorophyll. Wang et al. [16] found that the addition of DA considerably raised chlorophyll a and chlorophyll b under normal or stress conditions. The current outcomes assumed that DA application improved chl concentration by lowering ROS production (Figure 3) and hence maintaining a higher photosynthetic rate. Additionally, DA application increase N concentration in leaf tissue (Figure 2), which accelerates the synthesis of chlorophyll (Figure 2).

ND significantly decreased N absorption and accumulation; however, DA application under 100 or 50% RN enhanced the absorption and accumulation of N [3,5,34]. DA can stimulate nutrient uptake by modifying root architecture, including length, diameter, volume, surface area, and root hair numbers [34,43]. Moreover, DA facilitated N transport and partitioning under ND conditions. Nitrate reductase (NR) and nitrite reductase (NiR) are vital enzymes in the process of nitrate reduction [3,5,15]. Glutamine synthetase (GS) and glutamate synthase (GOGAT) are vital enzymes that elaborate on the process of ammonium assimilation [3,5,15,44]. The activity of N assimilation enzymes and transcriptional levels of the corresponding genes significantly decreased to a varying degree under ND [3,36], and exogenous DA noticeably improved the activity of decisive N metabolism enzymes under ND [3,45]. In this concern, Xiao-min et al. [5] found that Appel leaves under ND significantly decreased N% due to their declining effect on the activity of NR, NiR, GS, and GOGAT. Meanwhile, the promotive effect of DA on increasing N% resulted from their encouraging effect of DA on N-assimilation enzymes.

Nitrogen deficiency causes a disruption of cellular redox homeostasis and persuades the extra production of oxidative impairment biomarkers (Figure 3), which can upset cellular metabolism, cause direct mutilation to membrane lipids, and disturb the ability for ion uptake [9,46]. Equally, the initiation of cellular oxidative injury is a symbol of ND, which is designated by $H_2O_2$ and $O_2^{\bullet -}$ as well as cell death [47]. ND lessens the speed of the Calvin cycle and subsequently inhibits oxidation of NADPH and renewal of NADP1 and intensifies the Mehler reaction [1,48]. Among ROS, several considerations have been focused on $H_2O_2$, which has been deliberated as a major oxidant buildup with oxidative rupture. Peroxidation of membrane lipids is a crucial restriction that's used broadly for judging the degree of oxidative rupture and is predominantly due to boosting lipoxygenase activity [49]. Otherwise, the DA spraying for ND-influenced plants somewhat lessened the oxidative anxiety by deteriorating $O_2^{\bullet -}$, $H_2O_2$, and MDA assembly so avoiding ND damage. Current outcomes were supported by the study of Lan et al. [15], Jiao et al. [38], and Jodko-Piórecka et al. [50], who reported that DA application mitigated the harmful effects of oxidative stress. The DA possesses sufficient biochemical reactivates to elucidate its ability to simultaneously progress as a mediator of numerous physiological pathways and defense approaches [12]. Two potentialities might be predicted to elucidate the lowest oxidative injury noted in the DA-treated plants under ND. DA-treated plants suffered less cellular desiccation caused by a major desiccation-escaping approach [12]; that kept plants delimited alongside the assembly of ROS. Furthermore, DA usage protects plants from oxidative damage by scavenging the $O_2^{\bullet -}$ and activating antioxidant enzymes [3,14]. However, there is an increasing indication that DA attends as a chain-breaking antioxidant, imposing lipid peroxidation and adjusting antioxidant enzymes genes expression [15]. Such an impact would probably appearance the antioxidant property of DA for overwhelming the high ROS accumulation elicited by ND.

Crops possess a unique defense mechanism to eradicate ROS and preserve redox homeostasis in stressful situations [11]. The antioxidant system includes enzymatic and non-enzymatic antioxidants [9,10]. The antioxidant enzymes are typically reacting with

ROS and produce a harmless product [51]. A harmful by-product of oxidative stress known as superoxide radicals may interact with $H_2O_2$ to produce extremely reactive hydroxyl radicals. Superoxide dismutase converts hydroxyl radicals to $O_2$ and $H_2O_2$, which is then broken down by CAT and POD into molecular $O_2$ and $H_2O$ [51]. Therefore, the improvement of stress tolerance is frequently a result of improvements in the activity of antioxidant enzymes [7,14]. The current study found that DA application increased SOD, CAT, and POD activity over non-treated plants with 100% or 50% RN, allowing them to endure under ND conditions. Furthermore, some research has revealed that the application of DA considerably boosted the activity of antioxidant enzymes [14,52]. The impact is possibly partially ascribed to the antioxidant characteristics of DA and its derivative melanin, which is a powerful free radical eradication [12,14]. The antioxidant enzyme activity enhancement may result from either an adaptive modification in catalytic properties or the silent's gene transcription [53]. Usually, DA is able to achieve as (i) a direct ROS scavenger and (ii) an antioxidant system inducer to recover the antioxidant enzyme-encoding genes expression [50]. Also, DA treatments might motivate endogenous NO assimilation that can substitute for signaling molecules or ROS scavengers under stress features [54]. Therefore, it is realistic to conclude that DA protected crisphead lettuce plants from oxidative anxiety by preserving the greatest activity of antioxidant enzymes.

## 5. Conclusions

Current findings highlighted that exogenous application of 100 μM DA could reinforce the crisphead lettuce plant's resilience to ND by minimizing ROS accumulation and promoting enzymatic antioxidants alongside growth, yield, and quality improvement. Hence the current finding would validate extremely practical leafy vegetable growth under ND conditions. In addition, the study emphasizes the importance of minimizing the use of synthetic N fertilization to protect the environment while maintaining crop yield and quality. However, further research is necessary to fully understand the optimal concentrations and conditions for DA application to achieve maximum benefits.

**Author Contributions:** Conceptualization, S.F., M.A.M.A.E.-H., M.A.E.-S., K.H.A., M.M.H., E.F.A., S.A.A.-R. and H.A.E.-B.; methodology, S.F., M.A.M.A.E.-H., M.A.E.-S., K.H.A., S.A.A.-R., M.M.H., E.F.A. and H.A.E.-B.; software, S.F., M.A.M.A.E.-H., M.A.E.-S., K.H.A., S.A.A.-R., M.M.H., E.F.A. and H.A.E.-B.; validation, S.F., M.A.M.A.E.-H., M.A.E.-S., K.H.A., S.A.A.-R., M.M.H., E.F.A. and H.A.E.-B.; formal analysis, S.F., M.A.M.A.E.-H., M.A.E.-S., K.H.A., S.A.A.-R., M.M.H., E.F.A. and H.A.E.-B.; investigation, S.F., M.A.M.A.E.-H., M.A.E.-S., K.H.A., S.A.A.-R., M.M.H., E.F.A. and H.A.E.-B.; resources, S.F., M.A.M.A.E.-H., M.A.E.-S., K.H.A., S.A.A.-R., M.M.H., E.F.A. and H.A.E.-B.; data curation, S.F., M.A.M.A.E.-H., M.A.E.-S., K.H.A., M.M.H., E.F.A. and H.A.E.-B.; writing—original draft preparation, S.F., M.A.M.A.E.-H., M.A.E.-S., K.H.A., S.A.A.-R., M.M.H., E.F.A. and H.A.E.-B.; writing—review and editing, S.F., M.A.M.A.E.-H., M.A.E.-S., K.H.A., S.A.A.-R., M.M.H., E.F.A. and H.A.E.-B.; visualization, S.F., M.A.M.A.E.-H., M.A.E.-S., K.H.A., M.M.H., E.F.A. and H.A.E.-B.; supervision, S.F., M.A.M.A.E.-H., M.A.E.-S., K.H.A., S.A.A.-R., M.M.H., E.F.A. and H.A.E.-B.; project administration, S.F., M.A.M.A.E.-H., M.A.E.-S., K.H.A., M.M.H., E.F.A. and H.A.E.-B.; funding acquisition, S.F., M.A.M.A.E.-H., M.A.E.-S., K.H.A., S.A.A.-R., M.M.H., E.F.A. and H.A.E.-B. All authors have read and agreed to the published version of the manuscript.

**Funding:** This research was funded by the acknowledged Deanship of Scientific Research, Taif University.

**Data Availability Statement:** Not applicable.

**Acknowledgments:** The researchers would like to acknowledge the Deanship of Scientific Research, Taif University, for funding this work.

**Conflicts of Interest:** The authors declare no conflict of interest.

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
