# Peer review of "Effect of Dopamine on Growth, Some Biochemical Attributes, and the Yield of Crisphead Lettuce under Nitrogen Deficiency"

_horticulturae, doi:10.3390/horticulturae9080945_

Round 1

Reviewer 1 Report

An interesting manuscript which I enjoyed.

However, besides small additions to the methodology, I have a general concern about the use of principal component analysis. I consider its use for this data set to be absolutely nonsensical, unscientific, and unserious. there is a great deal of literature on the required sample size for principal component analysis. This should be considered and not a statistical procedure applied because it may be fashionable. In this manuscript there are 18 data sets (six treatments and three replicates) and 20 variables serving the PCA are included. The explanation of the variance with more than 99.9 % can be explained by the fact that there are no more degrees of freedom.

1. Please complete the missing methodological information in the attached manuscript.

2. Please remove all data related to the principal component analysis.

Up to the point 3.5 the evaluation is perfectly presented. also the correlation matrix is still interesting, especially regarding the quality characteristics and the vegetative growth and yield characteristics. The high correlation between the vegetative growth characteristics was of course to be expected.

If the PCA is omitted, the correlation coefficients can also be presented with two digits after the decimal point.

Author Response

Dear respective reviewer

 We appreciate the time and effort you have dedicated to providing insightful and helpful feedback on ways to strengthen our manuscript. We have incorporated changes that reflect the detailed suggestions you have graciously provided. We think that the manuscript has been greatly improved by these revisions and hope that our edits and the responses we provide below satisfactorily address all the issues and concerns you have noted. The necessary corrections have been written in The red lines in the manuscript. To facilitate your evaluation, the following is a point-by-point response to the questions and comments

An interesting manuscript which I enjoyed.

However, besides small additions to the methodology, I have a general concern about the use of principal component analysis. I consider its use for this data set to be absolutely nonsensical, unscientific, and unserious. there is a great deal of literature on the required sample size for principal component analysis. This should be considered and not a statistical procedure applied because it may be fashionable. In this manuscript there are 18 data sets (six treatments and three replicates) and 20 variables serving the PCA are included. The explanation of the variance with more than 99.9 % can be explained by the fact that there are no more degrees of freedom.

Thank you for your kind comment. We totally confirmed your suggestion, so as indicated in the revised manuscript, we deleted the PCA analysis

  1. Please complete the missing methodological information in the attached manuscript.

Thank you for your kind comment. We do all recommendations. As for

This is roughly equivalent to 30000 plants per hectare. Is this the usual number of plants per hectare in your region?

In Egypt, the number of plants per hectare ranged between 30000-32000 plants, so we used the density of 30000 plants per hectare.

Why only nine plants? According to which criteria were these plants taken?

As indicated in the revised manuscript line 136-137, we used 9 randomly selected plants and divided them into 3 replicates (each 3 plants represent one replicate) for each treatment.

To determine the leaf area of all leaves in head lettuce seems to me impossible, or very difficult. Please describe the procedure

We totally confirmed your comment. As indicated in the revised manuscript lines 142-145, we add the method of estimation of total leaf area according to Koller 1972

All leaves or only the outer ones as described later. Please define the procedure if it does not affect all leaves

Thank you, as indicated in the revised manuscript line 141 we used outer leaves.

For photosynthetic paper, all leaves or outer leaves?

Thank you, as indicated in the revised manuscript line 149 we used outer leaves

  1. 2. Please remove all data related to the principal component analysis.

Up to the point 3.5 the evaluation is perfectly presented. also the correlation matrix is still interesting, especially regarding the quality characteristics and the vegetative growth and yield characteristics. The high correlation between the vegetative growth characteristics was of course to be expected.

If the PCA is omitted, the correlation coefficients can also be presented with two digits after the decimal point.

Thank you for your kind comment. We totally confirmed your observation, so as indicated in the revised manuscript, we deleted the PCA analysis. We changed it

Once more, thank you for giving us the opportunity to strengthen our manuscript with your valuable comments and queries. We have worked hard to incorporate your feedback and hope that these revisions persuade you to accept our submission.

Thank you in advance for your time and attention.

Sincerely

Prof. Dr. Saad Farouk

Agric. Botany Dept.

Fac. of Agric.

Mansoura Univ.

Egypt

Reviewer 2 Report

The study introduced "Effect of dopamine on growth, some biochemical attributes, and yield of crisphead lettuce under nitrogen deficiency". The language is fine and the annual field trials were well developed and articulated. However, one-year date might not be sufficient to support the findings in a scientific sound. The baseline soil N values are relatively high so the plant response to N fertilization may be masked, which may lead to the failure in testing the objective. 

Fine

Author Response

Reviewer 2

Dear respective reviewer

 We appreciate the time and effort you have dedicated to providing insightful and helpful feedback on ways to strengthen our manuscript. We have incorporated changes that reflect the detailed suggestions you have graciously provided. We think that the manuscript has been greatly improved by these revisions and hope that our edits and the responses we provide below satisfactorily address all the issues and concerns you have noted. The necessary corrections have been written in The red lines in the manuscript. To facilitate your evaluation, the following is a point-by-point response to the questions and comments

The study introduced "Effect of dopamine on growth, some biochemical attributes, and yield of crisphead lettuce under nitrogen deficiency". The language is fine and the annual field trials were well developed and articulated. However, one-year date might not be sufficient to support the findings in a scientific sound. The baseline soil N values are relatively high so the plant response to N fertilization may be masked, which may lead to the failure in testing the objective. 

Thank you so much for your positive recommendation. As for the experimental period, we conduct a two-year experiment, the first one as a preliminary experiment using five nitrogen levels (100, 75, 50, 25, 10% from RN) with six DA spraying (0, 50, 100, 200, 400, 800 µM) on the total yield of crisphead lettuce. From the results of this experiment, we selected the exciting experimental treatments.

As for the baseline soil N values are relatively high. The content of total nitrogen (available and non-available) in the experimental field was extremely low as indicated by:

Z. LIANG; X. Q. ZHAO; X. Y. YI; Z. C. CHEN; X. Y. DONG; R. F. CHEN; R. F. SHEN (2013). Excessive application of nitrogen and phosphorus fertilizers induces soil acidification and phosphorus enrichment during vegetable production in Yangtze River Delta, China. Soil Use and Management, June 2013, 29, 161–168 doi: 10.1111/sum.12035

Zhang, Y.S., Ni, W.Z., Lin, X.Y., Lin, W.X. & Zhan, L.C. 1997. The state of the available nutrients of the vegetable garden soils in the suburb of Hangzhou and fertilizing countermeasures. In: Soil fertility in vegetable garden and rational fertilization of vegetables (eds J.C. Xie & J.X. Chen), pp. 43–46. Hehai University Press, Nanjing.

Once more, thank you for giving us the opportunity to strengthen our manuscript with your valuable comments and queries. We have worked hard to incorporate your feedback and hope that these revisions persuade you to accept our submission.

Thank you in advance for your time and attention.

Sincerely

Prof. Dr. Saad Farouk

Agric. Botany Dept.

Fac. of Agric.

Mansoura Univ.

Egypt

Reviewer 3 Report

View letter

1.       Please check whether the μM abbreviation is standardizedand check the expression and writing of all units in the full text.

2.       Line 48-52. Does this paragraph [1] only need to be used at the end

3.       Line 104, 2023 winter season

4.       Line 107-108superscript and subscript are wrong. The unit expression is wrong. The expression of 20.5 % N is not clear.

5.       Line 110-111The estimation of soil physical and chemical characteristics are not rigorous enough.

6.       Line 112check whether the full text decimal digits need to be unifiedTable1Please check the abbreviation of Organic matterOrganic matter is expressed by percentage or specific contentPlease check the unit of EC.

7.       Line 122. Unit expression is not standardized.

8.       Line 126-127The expression is wronghow to apply nitrogen fertilizer level

9.       Line 129,“at a rate of 60 units K2O ha-1. Please check whether the expression is correct.

10.    Line 135. The title and the text need to be separated, the text please start another line.

11.    From line 139please correct the indent problem of the first line.

12.    2.2.32.2.42.2.5. Three-level title format problem

13.    Line 175Whether the unit should be represented by abbreviationsLine 186The superscript is wrong, whether the unit should be expressed by abbreviation

14.    Figure 1 b. The error bar coincides with the letterFigure 1 c. Should the unit add parenthesesPlease check the full text of the figure

15.    Line 215Please check the correctness of the words.

16.    Line 224Figure(2 a-e)?

17.    Line 229237,“Figure 3, a-cFig. 3, a-c. The format needs to be unifiedplease check the full text similar format and unify.

18.    Line 232,“O2•−”,lack of standardization of writingplease check the full text and modify.

19.    Line 229Here is O2•−”,Figure 3 c. Why is Total chlorophyll”?

20.    Line 257-259,“confirm and might express conflict.

21.    Figure 4. Why add a Unit in all ordinates

22.    Figure 5 was written as Figure 4the order is wrong.

23.    Figure 6. Note format is not unifiedThere is an expression error in Figure Note ( b ) and the expression is not clearFigure a, b, c, d letters stackedneed to modify the font sizepay attention to the subscriptre-typesetting.

24.    Too much discussion text, please refine compression.

Author Response

Dear respective reviewer

 We appreciate the time and effort you have dedicated to providing insightful and helpful feedback on ways to strengthen our manuscript. We have incorporated changes that reflect the detailed suggestions you have graciously provided. We think that the manuscript has been greatly improved by these revisions and hope that our edits and the responses we provide below satisfactorily address all the issues and concerns you have noted. The necessary corrections have been written in The red lines in the manuscript. To facilitate your evaluation, the following is a point-by-point response to the questions and comments

  1. Please check whether the μM abbreviation is standardized and check the expression and writing of all units in the full text.

Thank you so much for your positive recommendation, as found in the revised manuscript we check all abbreviations in the whole manuscript

  1. Line 48-52. Does this paragraph [1] only need to be used at the end?

Thank you so much, as found in the revised manuscript we transferred this paragraph at the end of the introduction before the aim of the experiment (lines 87-98)

  1. Line 104, 2023 winter season?

Thank you so much, In Egypt, there are three main cultivation seasons, i.e. Winter season from September to April; the Nily season from April to June, and finally summer season from May to Sept.

  1. Line 107-108,superscript and subscript are wrong. The unit expression is wrong. The expression of 20.5 % N is not clear.

Thank you so much, we checked all manuscript

  1. Line 110-111,The estimation of soil physical and chemical characteristics are not rigorous enough.

Thank you so much, actually, we analyzed the soil’s chemical properties so deleted the physical characteristics of the experimental soil as indicated in lines 112-123

  1. Line 112,check whether the full text decimal digits need to be unified;Table1:Please check the abbreviation of Organic matter;Organic matter is expressed by percentage or specific content?Please check the unit of EC.

Thank you so much, actually, we checked and changed the Table units

  1. Line 122. Unit expression is not standardized.

Thank you so much, actually, we checked all units all over the manuscript like line 125

  1. Line 126-127,The expression is wrong,how to apply nitrogen fertilizer level?

Thank you so much, actually, we totally agree with your observation. We are rephrasing the sentence to be more understandable lines 129-131

  1. Line 129,“at a rate of 60 units K2O ha-1”. Please check whether the expression is correct.

Thank you so much, it is correct

  1. Line 135. The title and the text need to be separated, the text please start another line.

Thank you so much, we checked the whole manuscript

  1. From line 139,please correct the indent problem of the first line.

Thank you so much, we checked the whole manuscript

  1. 2.2.3,2.2.4,2.2.5. Three-level title format problem

Thank you so much, we checked the whole manuscript

  1. Line 175,Whether the unit should be represented by abbreviations?Line 186,The superscript is wrong, whether the unit should be expressed by abbreviation?

Thank you so much, we checked the whole manuscript and fixed the standardization or unity of units

  1. Figure 1 b. The error bar coincides with the letter;Figure 1 c. Should the unit add parentheses?Please check the full text of the figure

Thank you so much for your observation, we checked and correct all Figures as indicated in the revised manuscript

  1. Line 215,Please check the correctness of the words.

Thank you so much for your observation, we checked and correct the word in the title line 248

  1. Line 224,Figure(2 a-e)?

Thank you so much for your observation, we checked and corrected line 257

  1. Line 229,237,“Figure 3, a-c,Fig. 3, a-c”. The format needs to be unified,please check the full text similar format and unify.

Thank you so much for all your recommendations. We checked the whole manuscript and unify the format

  1. Line 232,“O2•−”,lack of standardization of writing,please check the full text and modify.

Thank you so much for all your recommendations. We checked the whole manuscript and unify the format

  1. Line 229,Here is “O2•−”,Figure 3 c. Why is “Total chlorophyll”?

Thank you so much for your observation, we checked and corrected Figure 3C page 8

  1. Line 257-259,“confirm” and “might” express conflict.

Thank you so much for your observation, we checked and corrected it to reveal line 292

  1. Figure 4. Why add a “Unit” in all ordinates?

Thank you so much for your observation, we checked and corrected it

  1. Figure 5 was written as Figure 4,the order is wrong.

Thank you so much for your observation, we checked and corrected it (Page 10)

  1. Figure 6. Note format is not unified;There is an expression error in Figure Note ( b ) and the expression is not clear;Figure a, b, c, d letters stacked,need to modify the font size;pay attention to the subscript;re-typesetting.

Thank you so much for your observation, we checked and corrected it page 11

  1. Too much discussion text, please refine compression.

Thank you so much for your recommendation, as indicated in the revised manuscript we checked the discussion section and shortened some sentences

Once more, thank you for giving us the opportunity to strengthen our manuscript with your valuable comments and queries. We have worked hard to incorporate your feedback and hope that these revisions persuade you to accept our submission.

Thank you in advance for your time and attention.

Sincerely

Prof. Dr. Saad Farouk

Agric. Botany Dept.

Fac. of Agric.

Mansoura Univ.

Egypt

Reviewer 4 Report

Chapter material and methods: the authors write line 110-111 that some physical and chemical properties have been estimated and cite them. and the table is described what chemical properties .... my question is why the authors did not study the properties of the soil on which they set up the experiment? why do they cite? in table 1, the notation of units (superscripts) should be corrected. then from line 121.. the authors write that they applied manure and mineral fertilization earlier. On what basis did they designate these doses if they did not test the soil themselves, they only cited citations? do the legal regulations in their country regarding the use of fertilizers (manure and mineral) provide any criteria/doses? I have no objections to the presentation and form of research results and discussions. Please expand the conclusions, write, among other things, the application significance of the conducted research.

The review was for an original research paper on The Effects of Dopamine....

Abstract spelled correctly. The introduction sufficiently introduces the research problem by moving the individual research aspects in turn.

Chapter material and methods. Please separate the individual subsections with a space to make them more readable.

line 154, reference 30 - isn't there an updated method?

Research results chapter. Figure 1 (e) replace the unit Ton - Mg.

The research results are presented very meticulously, extensively, supplemented with numerous graphics.

Discussion and conclusions written exhaustively.

In my opinion, the work is very extensive, the methodology, research results, discussion and conclusions are presented in detail and clearly enough.

I recommend for printing after literally minor corrections.

Author Response

Dear respective reviewer

 We appreciate the time and effort you have dedicated to providing insightful and helpful feedback on ways to strengthen our manuscript. We have incorporated changes that reflect the detailed suggestions you have graciously provided. We think that the manuscript has been greatly improved by these revisions and hope that our edits and the responses we provide below satisfactorily address all the issues and concerns you have noted. The necessary corrections have been written in The red lines in the manuscript. To facilitate your evaluation, the following is a point-by-point response to the questions and comments

Chapter material and methods: the authors write line 110-111 that some physical and chemical properties have been estimated and cite them. and the table is described what chemical properties .... my question is why the authors did not study the properties of the soil on which they set up the experiment? why do they cite? in table 1, the notation of units (superscripts) should be corrected.

Great thanks for your recommendation, we totally agree with your comment. As indicated in the revised manuscript we modify the text to be unity with text line 112

then from line 121.. the authors write that they applied manure and mineral fertilization earlier. On what basis did they designate these doses if they did not test the soil themselves, they only cited citations? do the legal regulations in their country regarding the use of fertilizers (manure and mineral) provide any criteria/doses?

Thank you so much for your revision, all fertilizer (mineral and organic) was added as a recommendation of the Ministry of Agriculture and Land Reclamation of Egypt

I have no objections to the presentation and form of research results and discussions. Please expand the conclusions, write, among other things, the application significance of the conducted research.

Great thanks for your positive encouragement and recommendation

 The review was for an original research paper on The Effects of Dopamine....

Abstract spelled correctly. The introduction sufficiently introduces the research problem by moving the individual research aspects in turn.

Chapter material and methods. Please separate the individual subsections with a space to make them more readable.

Great thanks for your positive encouragement and recommendation, we checked and modified the whole manuscript

line 154, reference 30 - isn't there an updated method?

Thank you, that is a prime reference for estimation of MDA, the other paper used it

Research results chapter. Figure 1 (e) replace the unit Ton - Mg.

Thank you, we changed

The research results are presented very meticulously, extensively, supplemented with numerous graphics.

Great thanks for your positive encouragement and recommendation

Discussion and conclusions written exhaustively.

Great thanks for your positive encouragement and recommendation

In my opinion, the work is very extensive, the methodology, research results, discussion and conclusions are presented in detail and clearly enough.

I recommend for printing after literally minor corrections.

Great thanks for your positive encouragement and recommendation

Once more, thank you for giving us the opportunity to strengthen our manuscript with your valuable comments and queries. We have worked hard to incorporate your feedback and hope that these revisions persuade you to accept our submission.

Thank you in advance for your time and attention.

Sincerely

Prof. Dr. Saad Farouk

Agric. Botany Dept.

Fac. of Agric.

Mansoura Univ.

Egypt

Round 2

Reviewer 2 Report

The quality of the manuscript has improved. Please add more details to the Statistical Analysis section. 

The language is fine. 

Author Response

Dear prospected reviewer 

We appreciate the time and effort you have dedicated to providing insightful feedback on ways to strengthen our manuscript. We have incorporated changes that reflect the detailed suggestions you have graciously provided. We think that the manuscript has been greatly improved by these revisions and hope that our edits and the responses we provide below satisfactorily address all the issues and concerns you have noted. The necessary corrections have been written in The red lines in the manuscript. To facilitate your evaluation, the following is a point-by-point response to the questions and comments

Thank you for your comments, I totally agree with your suggestions indicated in the attachment file

As for your recommendation, we do it in the revised manuscript in the section Statistical Analysis Page 5, lines 203-209. As “The data homogeneity was achieved formerly before the analysis of variance (ANOVA). All data were analyzed via a one-way ANOVA with the COSTATC statistical program (CoHort software, release 6.3.0.3, 2006; NC, USA) to test the impact of DA on moderating ND injury. The Tukey’s Honestly Significant Difference (HSD) test was used for comparing means at p ≤ 0.05. The means and standard deviation (SD) of three independent biological replications were utilized to present the data”

Additionally, we reformat the author’s contribution according to the journal requirement.

Once more, thank you for giving us the opportunity to strengthen our manuscript with your valuable comments and queries. We have worked hard to incorporate your feedback and hope that these revisions persuade you to accept our submission.

Thank you in advance for your time and attention.

Sincerely

Prof. Dr. Saad Farouk

Agric. Botany Dept.

Fac. of Agric.

Mansoura Univ.

Egypt

Reviewer 3 Report

View letter

1.       Line 106, 2023 winter has not yet arrived, ' 2022 / 2023 ' in ' 2023 ' should be removed.

2.       Table1. Please keep the last two digits of the decimal point, the full text is unified ; the abbreviation of ' Organic matter ' should be ' OM ' rather than ' O.M '.

3.       Line 144. The multiplication symbol is wrong.

4.       Line 148 and 180, the unit should be unified use of SI

5.       Line 261, 269, “Figure 3 a-cFig. 3 a-c” The format is not uniform.

6.       The number of the figure in Fig.5 appears two ' C ', please correct.

7.       Line 329. The principal component analysis is written in the figure 6, and the principal component analysis diagram is not seen.

8.       Line 412, 419, 428. The ' O2. ' format needs to be corrected in the discussion.

Minor editing of English language required

Author Response

Dear prospected reviewer 

We appreciate the time and effort you have dedicated to providing insightful feedback on ways to strengthen our manuscript. We have incorporated changes that reflect the detailed suggestions you have graciously provided. We think that the manuscript has been greatly improved by these revisions and hope that our edits and the responses we provide below satisfactorily address all the issues and concerns you have noted. The necessary corrections have been written in The red lines in the manuscript. To facilitate your evaluation, the following is a point-by-point response to the questions and comments

  1. Line 106, 2023 winter has not yet arrived, ' 2022 / 2023 ' in ' 2023 ' should be removed.

Thank you for your kind comment. As indicated in the revised manuscript P3, L106 we changed the presentation of this sentence.

  1. Table1. Please keep the last two digits of the decimal point, the full text is unified ; the abbreviation of ' Organic matter ' should be ' OM ' rather than ' O.M '.

Thank you for your kind comment. We keep the last two digits of the decimal point and changed the abbreviation of organic matter following your recommendation P3, L 115

  1. Line 144. The multiplication symbol is wrong.

Thank you for your kind comment. As indicated in P3 L 144 we changed it in a true symbol

  1. Line 148 and 180, the unit should be unified use of SI

Thank you for your kind comment. We checked all units used in all manuscript

  1. Line 261, 269, “Figure 3 a-c、Fig. 3 a-c” The format is not uniform.

Thank you for your kind comment. As indicated in the revised manuscript we unify the format all over the manuscript

  1. The number of the figure in Fig.5 appears two ' C ', please correct.。

Thank you for your kind comment. We checked and corrected P10

  1. Line 329. The principal component analysis is written in the figure 6, and the principal component analysis diagram is not seen.

Thank you for your kind comment. We checked and corrected the title of Figure  P11

  1. Line 412, 419, 428. The ' O2. − ' format needs to be corrected in the discussion.

Thank you for your kind comment. We checked and unify the abbreviation all over the manuscript

Once more, thank you for giving us the opportunity to strengthen our manuscript with your valuable comments and queries. We have worked hard to incorporate your feedback and hope that these revisions persuade you to accept our submission.

Thank you in advance for your time and attention.

Sincerely

Prof. Dr. Saad Farouk

Agric. Botany Dept.

Fac. of Agric.

Mansoura Univ.

Egypt
